# RGS5 Attenuates Baseline Activity of ERK1/2 and Promotes Growth Arrest of Vascular Smooth Muscle Cells

**DOI:** 10.3390/cells10071748

**Published:** 2021-07-11

**Authors:** Eda Demirel, Caroline Arnold, Jaspal Garg, Marius Andreas Jäger, Carsten Sticht, Rui Li, Hanna Kuk, Nina Wettschureck, Markus Hecker, Thomas Korff

**Affiliations:** 1Department of Cardiovascular Physiology, Institute of Physiology and Pathophysiology, Heidelberg University, 69120 Heidelberg, Germany; eda.demirel@bzh.uni-heidelberg.de (E.D.); caro.arnold@gmx.net (C.A.); jaspal.garg@physiologie.uni-heidelberg.de (J.G.); mariusjaeger@stud.uni-heidelberg.de (M.A.J.); hecker@physiologie.uni-heidelberg.de (M.H.); 2NGS Core Facility, Medical Faculty Mannheim, Heidelberg University, 68167 Mannheim, Germany; carsten.sticht@medma.uni-heidelberg.de; 3Department of Pharmacology, Max Planck Institute for Heart and Lung Research, 61231 Bad Nauheim, Germany; Rui.Li@mpi-bn.mpg.de (R.L.); Nina.Wettschureck@mpi-bn.mpg.de (N.W.); 4The Ottawa Department of Medicine, Faculty of Medicine, University of Ottawa, Ottawa, ON K1H 8M5, Canada; hkuk@uottawa.ca; 5European Center for Angioscience (ECAS), Medical Faculty Mannheim, Heidelberg University, 68167 Mannheim, Germany

**Keywords:** RGS proteins, G-protein signaling, VSMC phenotype, ERK1/2, blood pressure

## Abstract

The regulator of G-protein signaling 5 (RGS5) acts as an inhibitor of Gα_q/11_ and Gα_i/o_ activity in vascular smooth muscle cells (VSMCs), which regulate arterial tone and blood pressure. While RGS5 has been described as a crucial determinant regulating the VSMC responses during various vascular remodeling processes, its regulatory features in resting VSMCs and its impact on their phenotype are still under debate and were subject of this study. While *Rgs5* shows a variable expression in mouse arteries, neither global nor SMC-specific genetic ablation of *Rgs5* affected the baseline blood pressure yet elevated the phosphorylation level of the MAP kinase ERK1/2. Comparable results were obtained with 3D cultured resting VSMCs. In contrast, overexpression of RGS5 in 2D-cultured proliferating VSMCs promoted their resting state as evidenced by microarray-based expression profiling and attenuated the activity of Akt- and MAP kinase-related signaling cascades. Moreover, *RGS5* overexpression attenuated ERK1/2 phosphorylation, VSMC proliferation, and migration, which was mimicked by selectively inhibiting Gα_i/o_ but not Gα_q/11_ activity. Collectively, the heterogeneous expression of *Rgs5* suggests arterial blood vessel type-specific functions in mouse VSMCs. This comprises inhibition of acute agonist-induced Gα_q/11_/calcium release as well as the support of a resting VSMC phenotype with low ERK1/2 activity by suppressing the activity of Gα_i/o_.

## 1. Introduction

Vascular smooth muscle cells (VSMCs) form the contractile layer of the blood vessel wall and specifically respond to extra- and intracellular stimuli by increasing their tone to adapt blood flow and pressure. Stimuli by humoral agonists are usually sensed by corresponding receptors, which initiate signaling cascades to activate the contractile apparatus of the VSMCs. Heterotrimeric G-proteins—consisting of α subunits and the associated βγ-subunits—are important regulators of such signaling cascades determining vascular tone [1]. Their activation is usually initiated via binding of agonists to corresponding G-protein coupled receptors (GPCRs), which are capable to interact with one or several types of G-proteins including Gα_q/11_, Gα_12/13_, Gα_i/o_, and Gα_s_. In their inactive state, Gα-subunits are bound to guanosine diphosphate (GDP). Upon interaction with a ligand-stimulated GPCR, GDP becomes replaced by guanosine triphosphate (GTP) allowing the Gα subunits to interact and activate their specific target effector proteins [2]. For instance, while stimulation of Gα_q/11_ triggers the release of calcium from intracellular stores to promote acute contraction of VSMCs, Gα_s_-mediated signals have the opposite functional effect and may be counteracted by activation of Gα_i/o_. 

In addition to the agonist-induced VSMC contraction, G-protein signaling governs the activation of mitogen activated protein (MAP) kinases [3], which regulate several critical cellular functions, such as cell proliferation, differentiation, and apoptosis [4]. Upon activation, these kinases translocate from the cytoplasm to the nucleus and via phosphorylation control the activities of various transcription factors, subsequently mediating the above-mentioned cellular processes [5,6]. G-protein and MAP kinase signaling are connected via multiple pathways, including the activation of Ras via βγ-subunits [7], the activation of C-Raf by the Gα_q_-PKC axis and the enhancement of C-Raf activity by suppression of two, PKA and Rap1-dependent, inhibitory pathways via Gα_i_ [8]. Considering the relevance of these interconnected pathways for the control of cellular functions, the regulatory impact of G-protein-mediated signaling on acute and long-term responses of VSMCs to environmental stimuli is thus subject to further investigation. 

Although primarily triggered by agonists or environmental stimuli, the overall activity of the Gα-subunits and associated signaling events is controlled by regulators of G-protein signaling (RGS) [9,10]. Proteins of this family comprise at least 25 members with a characteristic GTPase-activating protein (GAP) domain that accelerates the hydrolysis of GTP bound to Gα subunits, promoting their inactive GDP-bound state and terminating the corresponding signal transduction [11,12]. As such, RGS proteins may shift the outcome of any G-protein-mediated signaling [13,14] and modulate acute responses of VSMCs to environmental stimuli as well as the baseline activity of individual signaling pathways. Based on previous findings, genetic ablation of *Rgs2* in mice prolonged preferentially Gα_q/11_-mediated signaling and caused hypertension [15] indicating its general relevance for the basal tone of VSMCs. In contrast, loss of RGS5—a potent inhibitor of Gα_q/11_ and Gα_i/o_ activity—resulted in hypo- [16], normo- [17], and hypertensive [18,19] mice depending on their genetic background and the applied methodology. Moreover, proliferation of VSMCs is controlled by RGS5 during neointima formation [20] and our own studies revealed the ability of RGS5 to stimulate RhoA activity in VSMCs in the context of arterial hypertension [21].

As can be deduced from the aforementioned findings, the regulatory influence of RGS5 on acute and baseline VSMC signaling, as well as its G-protein-specific regulatory features, are still under debate. In this context, we assumed that the reported partially contradicting functional features of RGS5 might be explained by context-specific inhibitory functions, which may promote diverging effects. Therefore, this study was intended to delineate long-term influence of RGS5 on different signaling pathways as well as on the functional status of resting and proliferating VSMCs by applying loss- and gain-of-function approaches. The observed effects were compared with those induced by selectively inhibiting the activity of Gα_q/11_ and Gα_i/o_ subunits to evaluate their individual contribution to the regulatory features of RGS5.

## 2. Materials and Methods

### 2.1. Antibodies and Reagents

For immunofluorescence studies, chicken anti-human/mouse RGS5 antibody was used (GW22900, Sigma Aldrich^®^, Schnelldorf, Germany). For capillary electrophoresis, the anti-human/mouse RGS5 antibody (BYT-ORB6873, Biozol, Eching, Germany) was utilized to detect RGS5. The rabbit anti-human Ki67 antibody used for quantification of cell proliferation was purchased from Abcam (ab16667, Cambridge, UK). The mouse anti-human/mouse/rat RhoA (26C4) antibody (sc-418) and the mouse anti-recombinant His-6 antibody (sc-8036) were purchased from Santa Cruz Biotechnology, Inc. (Heidelberg, Germany). The mouse anti-human VCP antibody (ab11433, Cambridge, UK) was purchased from Abcam. Antibodies against total ERK1/2 (#4695) as well as phosphorylated ERK1/2 (#4370) were purchased from Cell Signaling Technology^®^ (Danvers, MA, USA). Pertussis Toxin (#3097) was purchased from Tocris (Bristol, UK) and YM-25490 (#257-00631) from Fujifilm Wako Chemicals Europe GmbH (Neuss, Germany).

### 2.2. Generation of Inducible SMC-Specific Rgs5-Knockout Mice

Experiments in mice were carried out with prior permission from the Regional Council Karlsruhe (Permission Numbers: 35-9185.81/G-126/12; 35-9185.81/G-70/17) and conformed to the Guide for the Care and Use of Laboratory Animals (NIH). The *Rgs5*^−/−^ mouse strain (genetic background: F1H4 ((129S6/SvEvTac x C57BL/6NTac)F1) * C57BL/6) was backcrossed >15 generations onto the C57BL/6 background [17]. C57BL/6 wild type (WT) mice served as reference animals.

Mice of the line *Rgs5*^tm1a(EUCOMM)Wtsi^ (MGI ID:4432511, *Rgs5^fl/fl^*) were crossed with B6.FVB-Tg(Myh11-cre/ER^T2^)Soff/J mice (SM-MHC-CreER^T2^, kindly provided by Prof. Dr. Stefan Offermanns, Max Planck Institute for Heart and Lung Research, Bad Nauheim, Germany). The genotype of the offspring was assessed using the following primer sets:


*Rgs5^fl/fl^*


*Rgs5* LoxP F1: 5′-CACGTCCTTCTCTAGTTCGTG-3′

*Rgs5* loxP R2: 5′-GGAGTTCAAAACCATCCTCAG-3′

SM-MHC-CreER^T2^

SMWT1: 5′-TGACCCCATCTCTTCACTCC-3′

SMWT2: 5′-AACTCCACGACCACCTCATC-3′

phCREAS1: 5′-AGTCCCTCACATCCTCAGGTT-3′

In the corresponding adult male offspring, *Rgs5 was* genetically ablated by intraperitoneal application of 1 mg/kg/day tamoxifen (#T5648, Sigma-Aldrich, Schnelldorf, Germany) for five consecutive days (SMC-specific, tamoxifen-induced RGS5 knockout (*Rgs5^(SMC)−/)^*) or Miglyol (#3274, Caesar & Loretz GmbH, Hilden, Germany) as solvent control (*RGS5^fl/fl^*).

### 2.3. Telemetric Blood Pressure Measurement

Blood pressure measurements were performed in global- or SMC-specific *Rgs5*-deficient mice by implanting telemetric devices (PA-C10; Data Sciences International, St. Paul, MN, USA) as previously described [22]. In brief, through the left common carotid artery, catheter tips were advanced into the aortic arch of 12-week old mice. A signal transducer unit was placed in a subcutaneous pocket on the right ventrolateral side of the mouse. One week (recovery time) after surgery systolic/diastolic blood pressure measurements were recorded every 30 min for 5 min using Dataquest A.R.T. software 4.0.

### 2.4. Cell Culture 

The aorta of floxed *Rgs5* (*Rgs5^fl/fl^*) mice were used to isolate murine aortic SMCs (aoSMC). In brief, after careful removal of the surrounding adventitial tissue, the aorta was washed twice in Dulbecco’s PBS (without calcium and magnesium), cut in 1-mm–sized rings, and digested overnight with 1% collagenase (**#**C5138, Merck GmbH, Darmstadt, Germany). Suspended cells were seeded on culture plates, routinely checked for marker expression (e.g., αSMA, SMMHC), and used only until passage 5 throughout the experiments. The procedure for the human umbilical artery smooth muscle cell (HUASMC) isolation were carried out with the approval from the Local Ethical Committee (date: 12 April 2013/ID: S-191/2013, document number: 336/2005, Heidelberg, Germany) and conformed to the principles outlined in the Declaration of Helsinki (1997). HUASMCs were either isolated from human umbilical cord arteries or purchased from ProVitro (Berlin, Germany) and were cultured up to passage 5. All VSMCs were cultured in low-glucose DMEM (Thermo Fisher Scientific, Waltham, MA, USA) supplemented with 15% fetal calf serum (Biochrom, Cambridge, UK).

### 2.5. siRNA-Based Gene Silencing

To knockdown RGS5 in HUASMCs, siRNA-based silencing approach was utilized. HUASMCs were transfected with 100 nM of either Control siRNA (Dharmacon™ siGenome, control siRNA D-001210-01-20) or *RGS5*-targeting siRNA (sense: CCUGAAGUCUGAAUUCAGU, antisense: CCAUGAAUGUGGACUGGCA, purchased from Sigma Aldrich) using the MATRAsi technique (IBA Lifesciences, Göttingen, Germany) by following manufacturer’s instructions. After 48 h of incubation of transfected cells for, the efficiency of RGS5 knockdown was verified using qPCR analysis (Appendix A).

### 2.6. Adenoviral Transfection

HUASMCs were transduced using adenoviral vectors (MOI 200, kindly provided by Prof. Dr. Thomas Wieland, Mannheim, Germany), which have been described earlier [17] to overexpress only GFP as a control (Ad-GFP) or His-tagged RGS5 and GFP (Ad-RGS5) both under a CMV promoter. Cells were transduced at 70% confluency in medium supplemented with 2 µg/mL polybrene^®^ (sc-134220, Santa Cruz Biotechnology, Inc., Heidelberg, Germany) to improve transfection efficiency. Cells were washed after 18 h to remove the viruses.

Murine SMCs (aoSMCs) were transduced with adenoviruses (MOI 1000) carrying empty vector (Ad-CMV-Null, #1300) or Cre-recombinase (Ad-CMV-iCRE, #1045N) both purchased from Vector Biolabs, Malvern, PA, USA to delete the *Rgs5*. mRNA analysis was performed 72 h after transduction to verify the genetic ablation of *Rgs5*. Transduced-cells were further used for generating three-dimensional (3D) spheroids.

### 2.7. Generation of 3D Spheroids

3D cellular-spheroids were generated using hanging drops methodology. Briefly, cells were detached with trypsin, centrifuged at 300× *g* for 5 min, resuspended and counted in a Neubauer chamber. A total of 3000 (HUASMC) or 500 cells (aoSMC) were mixed per 25 μL medium with 0.24% *w*/*v* methyl cellulose (#M0650, Merck, Darmstadt, Germany) and 15% FCS and spotted dropwise onto flat petri dishes. Petri dishes were turned upside down and incubated for 48 h to generate hanging drops. Petri dishes were washed with 10 mL of DPBS (#14040091, Thermo-Fisher, Waltham, MA, USA) to harvest the 3D spheroids, which were centrifuged at 300× *g* for 5 min and then processed for mRNA or protein analyses.

### 2.8. 3D Migration Assay

The 3D spheroids (500 cells/spheroids) were generated by hanging drop method for 24 h and subsequently embedded into collagen matrices. A total of 4.5 mL acidic collagen extract of rat tails (5 mg/mL, 4 °C) was mixed with 500 μL of 10× M199 (#M0605, Sigma-Aldrich), and titrated with 0.2 M NaOH (approximately 500 μL) to neutralize the stock solution. This stock solution was then mixed with equal volume of DMEM (containing 30% FCS, 1.2% w/v methyl cellulose) and the 3D spheroids (about 75 spheroids/mL). This mixture (1 mL/well) was transferred into pre-warmed 24-well plates and incubated at standard conditions for 30 min. Test substances dissolved in 0.1 mL DMEM/ 15% FCS were pipetted on top of each well. After 24 h, the cumulative length of the sprouts originating from one spheroid was determined using an Olympus CKX 40 microscope (Olympus CACh 10x/0.2 NA dry objective lens, Olympus U-LS03-3 camera) and the imaging software CellD (v.3.4; Olympus, Hamburg, Germany). For each experimental group, 10 spheroids were analyzed and an average cumulative sprout length per spheroid was determined.

### 2.9. Microarray-Based Transcriptome Analysis

HUASMCs were transduced with adenoviruses encoding GFP or RGS5 as described. Total RNA from these cells was isolated using the RNeasy Mini Kit (#74106, Qiagen, Hilden, Germany). Gene expression profiling was performed using arrays of human HuGene-2.0-st-type (#902112, Thermo Fisher Scientific, Waltham, MA, USA). Using Affymetrix standard labelling protocol utilizing the GeneChip WT Plus Reagent Kit and the GeneChip Hybridization, Wash and Stain Kit (Affymetrix, Santa Clara, CA, USA), biotinylated antisense cDNA was prepared. Hybridization on the chip was carried out in a GeneChip Hybridization oven 640 (Affymetrix), followed by dyeing in the GeneChip Fluidics Station 450 (Affymetrix) and thereafter scanned with a GeneChip Scanner 3000 (AffymetrixA custom Chip Design File (v.22, CDF)) with Entrez-based gene definitions was used to annotate the arrays [23]. The raw fluorescence intensity values were normalized applying quantile normalization and RMA background correction. One-way ANOVA was performed to identify differentially expressed genes using a commercial software package JMP10 Genomics v.6 from SAS (SAS Institute, Cary, NC, USA). A false positive rate of a = 0.05 with false-discovery rate correction was taken as the level of significance. Gene Set Enrichment Analysis (GSEA) was performed to determine whether defined lists (or sets) of genes exhibit a statistically significant bias in their distribution within a ranked gene list. The statistical procedure described was performed using the software GSEA. Pathways belonging to various cell functions, such as cell cycle or apoptosis, were obtained from public external databases (KEGG, http://www.genome.jp/kegg, accessed on August 2019).

### 2.10. Quantitative Real Time RT-PCR (qPCR) Analysis

Murine arteries were isolated and frozen in QIAzol Lysis reagent (#79306, Qiagen, Hilden, Germany). The tissue was disrupted using a tissue lyser (Retsch, MM301, 30 Hz, 1.5 min) and 5 mm stainless steel beads. For RNA isolation from cells, RLT Buffer (#79216, Qiagen) containing 1% β-Mercaptoethanol was used to lyse the cells. Total RNA was isolated by solid-phase extraction using the RNeasy^®^ Micro Kit (#74004, Qiagen) by following the manufacturer’s instructions. Subsequently, cDNA was synthesized using the Sensiscript Reverse Transcription Kit (#205213, Qiagen). Quantitative real-time RT-PCR for the target sequences (primer sequences are listed in Table 1) was performed in the Rotor-Gene Q (Qiagen) using the LightCyclerR 480 SYBR Green I Master Mix (#04707516001, Roche, Mannheim, Germany). Fluorescence was monitored (excitation at 470 nm and emission at 530 nm) at the end of the annealing phase. Threshold cycle (Ct) was set within the exponential phase of the PCR. Quantification of the PCR product was done by using the ΔΔCt method. Amplification of the 60S ribosomal protein L32 (RPL32) cDNA served as an internal reference.

### 2.11. Electrophoresis-Based Protein Analyses

Cells were lysed in RIPA buffer (65 mM Tris, 154 mM NaCl, 10% sodium deoxy-cholate, 10% NP-40, 1 mM EDTA, pH 7.4), and centrifuged for 15 min at 13,000× *g* at 4 °C. Clear supernatant was boiled at 95 °C for 10 min in sample buffer (2 mL glycerol, 2 mL 1 M Tris (pH 6.8), 1 mL 2-ME, 2 mg bromophenol blue, and 400 mg SDS). Proteins were separated on 12% SDS-PAGE gel, blotted onto a PVDF membrane, and analyzed by ECL-based immuno-detection following manufacturer’s instructions. Primary antibodies in following dilutions were used: rabbit anti- human/mouse pERK1/2, 1:1000; rabbit anti-human/mouse ERK1/2, 1:1000; mouse anti-human/mouse/rat RhoA, 1:500; mouse anti-His6, 1:1000. ImageJ (v.1.49a; NIH) software was used to quantify gray intensities. Protein levels were calculated relative to appropriate loading controls—e.g., total ERK1/2 for pERK1/2 and representative bands were always shown from the same membrane.

Capillary electrophoresis (Figure 2, Figure 3 and Figure 6e, Appendix A) was used for separating and automated detection of proteins in RIPA lysates employing the WES™ system (ProteinSimple, San Jose, CA, USA). Reagents and samples were prepared according to manufacturer’s instructions. After dilution of samples to a final concentration of 1 mg/mL, protein separation, and detection steps were carried out with the following settings: separation voltage, 375 V; separation time, 30 min; Antibody diluent time, 5 min; primary antibody time, 90 min; and secondary antibody time, 30 min. For capillary electrophoresis, following dilutions of primary antibodies were used: VCP, 1:50; RGS5, 1:25; ERK1/2, 1:50; and pERK1/2, 1:50. Signal intensities of bands were automatically analyzed using the Compass SW software (v3.1.7; ProteinSimple) and, protein levels were reported relative to appropriate loading controls.

### 2.12. Human Phosphokinase Array

Proteome Profiler Human Phospho-Kinase Array Kit (#ARY003B, R&D Systems^®^, Minneapolis, MN, USA) was utilized to simultaneously determine the relative phosphorylation levels of several kinases in the cell lysates by following manufacturer’s instructions. Briefly, cell lysates were incubated with the membranes where specific phospho-specific primary antibodies have been spotted. After washing steps, membranes were incubated with biotinylated detection antibodies followed by Streptavidin-HRP. Chemiluminescent detection was carried out using the Image Quant™ TM LAS 4000 mini (GE Healthcare, Waukesha, WI, USA). Grey intensities were quantified using the ImageJ Software (NIH, USA, Version 1.49a).

### 2.13. Whole Mount Immunofluorescence Analysis

Mouse arteries were washed in PBS, glued (acryl tissue glue) onto a microscope slide and fixed in Dent’s fixative. After 10 min of consecutive rehydration steps in 75% methanol/PBS, 50% methanol/PBS, 25% methanol/PBS, arteries were washed in PBS-TD (1% Tween^®^ 20, 1% DMSO) and incubated at room temperature in blocking solution (0.1 M glycine, 0.5% casein, 2% goat serum in PBS-TD) for 1 h. The primary chicken anti-human/mouse RGS5 antibody (1:200) was incubated overnight at 4 °C. After washing the arteries 6 times in PBS-TD, the secondary goat anti-chicken-Cy3 antibody (#103-165-155, 1:500, Dianova GmbH, Hamburg, Germany) was incubated overnight at 4 °C. After further washing 6 times with PBS-TD, the arteries were stained with DAPI (1:5000, in PBS-TD) and mounted in Mowiol (Calbiochem, San Diego, CA, USA). An IX81 confocal (spinning disk) microscope (Olympus, Hamburg, Germany) was used to acquire images of all tissue layers to generate a Z-stack of images, which were presented as maximum intensity (fluorescence) projection by applying the Olympus Xcellence software (version 2.0, Build 4768). 

### 2.14. Measurement of Intracellular Calcium Mobilization

Cells or spheroids were incubated with 2.5 μM Rhodamine-4 AM (AAT BioQuest, Sunnyvale, CA, USA) dissolved in HBSS Buffer with Ca^2+^/Mg^2+^ (PAA, Pasching, Austria) containing 20 mM HEPES and 0.001% detergent (Pluronic F12, AAT BioQuest) for 30 min at 37 °C and followed by another 30 min at RT in the dark. Angiotensin II (#A6402, Sigma) and Norepinephrine (#A0937, Sigma)-evoked mobilization of intracellular calcium was measured using the fmax Fluorimeter (Molecular Devices, Biberach, Germany). Relative fluorescence units were measured every 5 s for a period of 5 min and were normalized to non-stimulated controls.

Fluorescence intensity of spheroids was determined by live cell fluorescence imaging. To this end, spheroids were transferred to 96 well plates, incubated at 37 °C, 5% CO_2_ in a Tokai Hit INU incubation unit, and stimulated with the indicated agonists. The fluorescence intensity of the spheroid was recorded every second by a Hamamatsu ORCA-R2 C10600-10B camera and quantified by utilizing the cellSens dimension software (v.1.12; Olympus).

### 2.15. RhoA Activity Assay

GTP-bound fraction of RhoA was measured, using the G-LISA RhoA Activation Assay Biochem Kit (Cytoskeleton, Inc., TebuBio, Offenbach, Germany). HUASMCs were cultured in 6-well plates and, subsequently, transfected with either control siRNA or *RGS5*-targeting siRNA to deplete RGS5 Cells, were lysed, and the protein concentration was determined and adjusted for further analyses according to the manufacturer’s instructions. GTP-bound RhoA was normalized to the total RhoA content of the samples, which was assessed by Western blot analysis.

### 2.16. Immunofluorescence-Based Detection of Ki67

HUASMCs were washed with HBSS, and fixed with 100% methanol (−20 °C) for 15 min and dried. Fixed cells were blocked for 1 h in 0.25% casein and incubated with the anti-Ki67 antibody (1:2500) at 4 °C overnight. After three washing steps, cells were incubated with the secondary antibody for 1 h at RT. After staining of the nuclei with DAPI, the fluorescence was recorded using Olympus IX83-System with cellSens Software (v. 1.12). The percentage of Ki67-positive nuclei was quantified by utilizing the ImageJ Software (v 1.49).

### 2.17. Statistical Analysis

All results are reported as means ± SD and statistically analyzed by utilizing GraphPad Prism (Version 9.1). Outliers were identified by application of the Grubbs’ test with α set to 0.05. Differences among normally distributed values of two matched experimental groups were analyzed by unpaired Student’s *t*-test or one sample *t*-test if appropriate. *p* < 0.05 was defined as statistically significant. Differences of one parameter between normally distributed values of three or more experimental groups were analyzed by one-way ANOVA followed by Šídák’s multiple comparisons test for selected pairs of groups or a Tukey–Kramer test for multiple comparisons. *p* < 0.05 was considered statistically significant (* *p* < 0.05, ** *p* < 0.01, *** *p* < 0.001).

## 3. Results

### 3.1. Regulatory Features of RGS5

RGS5 inhibits the activity of both Gα_q/11_ and Gα_i/o_-dependent signaling pathways and, thus, may decrease the tone of VSMCs by simultaneously attenuating Gα_q/11_-PLC-induced IP3/calcium- and DAG/PKC-mediated contraction, and by limiting the inhibition of Gα_i/o_-adenylyl cyclase-cAMP-PKA-mediated relaxation. Moreover, the inhibition of these signaling pathways may also affect the activity of RhoA- and mitogen activated (MAP) kinases, which are connected to the G-protein activity via βγ-subunits, PKA and PKC [8]. While these regulatory features illustrate the fundamental capacity of RGS5 for controlling VSMC functions, the actual inhibitory impact may be dependent on its context- and cell-specific expression.

#### 3.1.1. Heterogeneous Artery Type-Specific Expression of Rgs5 in Mice

Considering the diverse developmental origin of different VSMC populations associated with varying levels of *Rgs5* expression [24], we assumed artery type-specific differences in the *Rgs5* expression of mouse arteries. Corresponding analyses of different mouse arteries included small arteries, such as third order branches of the mesenteric artery and large conduit arteries such as the aorta. These exhibited marked differences in the levels of *Rgs5* expression (Figure 1a), which appeared to correlate, at least in part, with the detectable protein levels (Figure 1b). In contrast, the expression levels of *Rgs2* and *Rgs16* were much lower and homogeneous as compared to *Rgs5* (Figure 1a).

#### 3.1.2. Ablation of *Rgs5* Does Not Alter the Systolic and Diastolic Blood Pressure Values but Stimulates ERK1/2 Phosphorylation

Based on these findings, we investigated whether loss of *Rgs5* affects the arterial baseline tone in mice. We indirectly assessed this parameter by performing a continuous telemetry-based analysis of the blood pressure in wild type (WT) and *Rgs5*-deficient (*Rgs5^−/−^*) mice. No differences were detected for the averaged systolic and diastolic blood pressure values (Figure 2a). Similar results were obtained with mice upon SMC-specific ablation of *Rgs5* (Figure 2b, Appendix A), indicating that RGS5-mediated effects in other cell types (e.g., endothelial cells) may exert a negligible impact on the general vascular tone.

Considering that knockout of *Rgs2*—another inhibitor of Gα_q/11_ and Gα_i/o_-signaling—causes an increase in baseline blood pressure [15], these results suggest an artery-type specific functional role of RGS5. We assumed that arteries showing low expression of *Rgs5* such as branches of the mesenteric artery were less affected by its knockout than arteries with robust *Rgs5* expression such as large conduit arteries. Consequently, we selected the caudal artery and aorta to investigate whether loss of *Rgs5* may influence cellular signaling cascades. We focused on the MAP kinases ERK1/2 as their activity is known to be modulated by G-proteins [8] as well as RGS5 [25] and found an increased level of ERK1/2 phosphorylation after SMC-specific knockout of *Rgs5* (Figure 2c). 

#### 3.1.3. High Levels of RGS5 Are Maintained in Mouse and Human VSMCs under 3D Culture Conditions In Vitro

For further mechanistic analyses addressing the consequences of *Rgs5* deficiency, we required an experimental setup that maintains *Rgs5* expression in cultured mouse aortic VSMCs. We applied a 3D spheroid culture technique that allows the formation of multiple SMC layers and promotes a differentiated, contractile and quiescent VSMC phenotype [26]. As compared to proliferating murine VSMCs cultured under 2D conditions, *Rgs5* expression was upregulated in 3D spheroids (Figure 3a) while its knockout (Figure 3b) was associated with increased ERK1/2 phosphorylation (Figure 3c). Comparable results were obtained with human VSMCs maintaining high levels of RGS5 in 3D spheroids (Figure 3d,e) and significantly reduced ERK1/2 phosphorylation (Figure 3f) as compared to VSMCs cultured as 2D monolayer.

#### 3.1.4. Knockdown of RGS5 in Human VSMC Spheroids Inhibits RhoA Activity and Augments Agonist-Induced Calcium Release

For further validation of this cell culture model, we tested if knockdown of *Rgs5* (i) inhibits the activity of RhoA in VSMCs as has been observed in vivo [21] and (ii) amplifies the G-protein-mediated release of calcium. Culturing of VSMCs in 3D spheroids increased RhoA activity as compared to 2D cell culture, while siRNA-mediated knockdown of *Rgs5* expression (Appendix A) decreased the activity of RhoA as evidenced by determining the level of RhoA-GTP (Figure 4a). Intracellular calcium transients were assessed in VSMC spheroids under baseline conditions and upon stimulation with GPCR agonists. Knockdown of *Rgs5* increased the intracellular baseline calcium levels (Figure 4b) and amplified the transient calcium release upon stimulation with angiotensin II or norepinephrine (Figure 4c–e).

#### 3.1.5. Overexpression of *Rgs5* and Inhibition of Gα_i/o_ Signaling Inhibits ERK1/2 Activity, Proliferation and Migration of Human VSMCs

The aforementioned findings suggested that in resting VSMCs, high levels of RGS5 may functionally contribute to cellular quiescence and differentiation e.g., by attenuating the impact of stimulating agonists and supporting RhoA activity [27] while limiting the activity of MAP kinases. Elevated expression of *Rgs5* in active and proliferating VSMCs may thus in turn force their quiescent state. To test this hypothesis, we overexpressed RGS5 in 2D cultured proliferating VSMCs (Appendix A) and investigated alterations of their transcriptome by applying whole genome microarray analyses (Appendix A). Based on the outcomes of the microarray analyses, RGS5 decreased the level of transcripts associated with DNA replication and cell cycle control (Figure 5a,c) including *CCNE2, CDC6,* and *Ki67* (Figure 5b) and stimulated the expression of genes encoding cytokines or molecular components controlling cell adhesion and interaction with the extracellular matrix (Figure 5a,b,d). In addition, RGS5 also attenuated the expression of genes associated with the metabolism (butanoate, purine, pyrimidine), as well as translation and RNA processing, suggesting a general decline in cellular activity. Among the top upregulated transcripts were *KCNK2*—encoding a mechanosensitive potassium channel [28], and MIR145—a key mediator of smooth muscle cell differentiation [29]. However, the gene set attributed to “vascular smooth muscle contraction” was not significantly altered (NES: 1.17, adj. p-val. 0.41). Likewise, no significant changes in expression were observed for prototypic smooth muscle cell differentiation marker genes such as *MYH11*, *ACTA2*, *CNN1,* or *TAGLN*.

On the signaling level, *Rgs5* overexpression decreased the phosphorylation of several protein kinases, such as Akt (Figure 6a) and the MAP kinases ERK1/2 and JNK (Figure 6b), which was determined by quantitating their phosphorylation of activating phosphorylation sites. Correspondingly, RGS5 elevated the phosphorylation level of CREB and c-Jun—targets of Akt and MAP kinases, respectively. 

We next investigated which G-protein-dependent signaling pathway may mediate the observed responses of VSMCs by selective inhibition of either Gα_i/o_ or Gα_q/11_ activity. Pertussis toxin (PTX)-mediated inhibition of Gα_i/o_, but not YM254890-based interference with Gα_q/11_ activity (Appendix A) decreased ERK1/2 phosphorylation (Figure 6c–e). In addition, proliferation and migration of VSMCs were analyzed as functional readouts representing the activated VSMC phenotype. Immunofluorescence-based detection of the proliferation marker Ki67 in the nuclei of VSMCs indicated an inhibitory influence of RGS5 on the cell cycle (Figure 7a). Furthermore, RGS5 mitigated the outgrowth of VSMCs from VSMC-spheroids into a 3D collagen matrix (Figure 7b). Again, both inhibitory effects were mimicked by interfering with the activity of Gα_i/o_ but not Gα_q/11_ (Figure 7c,d) or by stimulating adenylyl cyclase through forskolin (Appendix A).

## 4. Discussion

The phenotype and function of VSMCs is defined by a plethora of developmental and environmental determinants (e.g., blood pressure). As such, arterial VSMCs of the adult vasculature represent a phenotypically diverse cell population with organ-specific functional features. The diversity of the VSMC phenotype was recently highlighted by single cell RNA-sequencing analyses, which revealed that the gene expression and protein pattern of GPCRs is highly diverse even within a population of VSMCs from the same type of artery [30]. For instance, heterogeneous expression was observed for most GPCR transcripts in both large caliber conductance and small caliber resistance arteries, while some GPCRs, such as *Ednrb* (endothelin receptor type-B) appeared preferentially expressed in the latter. Given the complexity of the resulting signaling scenario for each VSMC subpopulation, the relevance of RGS proteins for balancing the activity of individual signaling pathways becomes even more evident. To this end, our results indicate an artery subtype-specific expression of RGS5 thereby verifying earlier reports suggesting that the developmental origin of VSMCs determines the RGS5 expression level [24]. In line with our observations, in silico analyses utilizing a database for tissue-specific gene expression [31] revealed highly heterogeneous expression levels of *RGS5* in human arteries (Appendix A). Moreover, single cell RNAseq-based data [32] from mice also suggests diverse expression levels of *Rgs5* even within VSMC populations from one organ. For instance, *Rgs5* is strongly expressed in venous and arteriolar VSMCs of the brain but barely detectable in arterial VSMCs and endothelial cell (EC) types (Appendix A). Such findings also imply a minor relevance of RGS5 for EC signaling in mice, at least under baseline conditions. Correspondingly, our study for the first time shows comparable arterial baseline blood pressure values in mice with SMC-specific and systemic knockout of *Rgs5,* suggesting that additional loss of endothelial RGS5 in the latter mouse line does not affect blood pressure regulation. Moreover, our results also imply that genetic ablation of *Rgs5* evokes an artery subtype-specific effect rather than a general lapse of systemic blood pressure regulation.

With respect to mechanisms controlling the expression of *Rgs5* in mice, several molecular and environmental determinants have been identified. On the epigenetic level, methylation of its promoter may control *Rgs5* expression in a developmental origin-specific manner [24]. Arterial hypertension appears to downregulate *Rgs5* expression in angiotensin II-treated mice [18] while the opposite was observed in deoxycorticosterone acetate (DOCA)-treated hypertensive mice [21] indicating a context-dependent regulation. On the cellular level, *Rgs5* expression was increased in VSMCs or arteries exposed to nitric oxide or biomechanical stretch [17,21] and repressed in VSMCs under mitogenic culture conditions [20] or after treatment with platelet-derived growth factor [33]. Here, we show that the low expression level of *RGS5* in proliferating VSMCs is robustly increased under 3D culture conditions supporting the resting VSMC phenotype [26]. In this context, RGS5 appears to act as a determinant of cellular quiescence as it attenuates Gα_q/11_ and Gα_i/o_-protein activity, and significantly alters the activity of downstream signaling cascades. For instance, RGS5 promotes the baseline activity of RhoA confirming earlier observations and its link with Gα_i/o_ and Gα_q/11_—mediated signaling events [21]. Functionally, RhoA activity stimulates the expression of prototypic SMC markers, such as αSMA and SM22 [27]—a hallmark of a contractile and quiescent VSMC phenotype that predominates under the chosen 3D culture conditions [26]. Moreover, RhoA may also contribute to transforming growth factor-β-dependent Smad signaling, which supports VSMC differentiation [34,35] and is critical for preserving the growth arrest in 3D VSMC spheroids [26].

Inhibitory features of RGS5 described within this study apply to both, the acute agonist-induced stimulation of Gα_q/11_-dependent calcium release from intracellular stores and the baseline activity of the MAP kinase ERK1/2. The former observation relies on the well-described termination of Gα_q/11_-mediated calcium mobilization [13,17], which may also affect the recently described Gα_q/11_-restricted Gα_i_-Gβγ-PLCβ-mediated calcium release [36]. Likewise, a principal link of ERK1/2 phosphorylation and G-protein signaling has been repeatedly reported for different experimental conditions and cell types [13,15,31,32]. Several G-protein-associated signaling sources may be involved such as the Gβγ-Ras axis [3], the Gα_q/11_-PKC axis [37,38] and, though in an inverse fashion, also the Gα_i/o_-PKA axis considering that PKA inhibits Gα_q/11_-mediated signaling cascades [39,40,41]. Interestingly, the inhibitory effects of RGS5 on the acute agonist-induced calcium release is abolished upon selective inhibition of Gα_q/11_ while baseline MAP kinase activity is only blocked by selective inhibition of Gα_i/o_ but not Gα_q/11_. This implies as yet unrecognized agonist-independent G-protein-associated MAP kinase activity in mouse and human VSMCs that is mainly balanced via Gα_i/o_-dependent signaling pathways. Thus, suppression of Gα_i/o_ activity through RGS5 would enhance the adenylyl cyclase-cAMP-PKA signal strength with the capacity to limit MAP kinase activation, as was evidenced by the forskolin-induced inhibition of ERK1/2 phosphorylation. Additionally, RGS5 would also abrogate Gα_q/11_-PKC- as well as Gβγ-dependent MAP kinase activation since termination of Gα activity is known to preclude the dissociation of the corresponding Gβγ-subunits. However, the overall impact on ERK1/2 phosphorylation brought about by functionally inhibited Gα_q/11_ was minor. This may also provide a possible explanation why selective inhibition of Gα_q/11_ potently interferes with the directly linked agonist-induced calcium release but not with baseline MAP kinase activity as an indirectly connected downstream signaling event that is controlled by many additional signaling cascades. In fact, restricting Gα_i/o_ activity has a much greater relevance in this context.

On the functional level, RGS5 has been reported to control many aspects of the VSMC phenotype. For instance, its overexpression inhibits VSMC proliferation during neointima formation in mice [20]. Comparable results were obtained in this study whereby overexpression of RGS5 inhibited proliferation and migration of cultured human VSMCs. However, phenotype changes of VSMCs evoked by biomechanical stretch or wall stress as it occurs during hypertension-induced structural remodeling of the arterial wall were impaired in mice upon knockdown of *Rgs5* [21]. It may thus be tempting to speculate that chronically amplified MAP kinase activity caused by the loss of *RGS5,* as we have observed it in vitro and in vivo, may indeed promote the activated and synthetic phenotype of VSMCs. In the long run, this maladjustment may support thickening and stiffening of the media. Considering the artery type-specific expression of *RGS5,* the relevance of such remodeling processes for individual sites of the arterial system may nevertheless be highly diverse.

## 5. Conclusions

Collectively, RGS5 acts as a context-sensitive artery type-specific regulator that adjusts acute and long-term G-protein-mediated signaling in VSMCs. The inhibition of Gα_i/o_ and Gα_q/11_-dependent signaling caused by its expression supports the resting phenotype of VSMCs by silencing MAP kinase activity, likely by controlling Gα_i/o_-dependent signaling pathways. In contrast, SMC-specific loss of RGS5 stimulates baseline MAP kinase activity of arteries without affecting the systemic arterial blood pressure.

## Figures and Tables

**Figure 1 cells-10-01748-f001:**
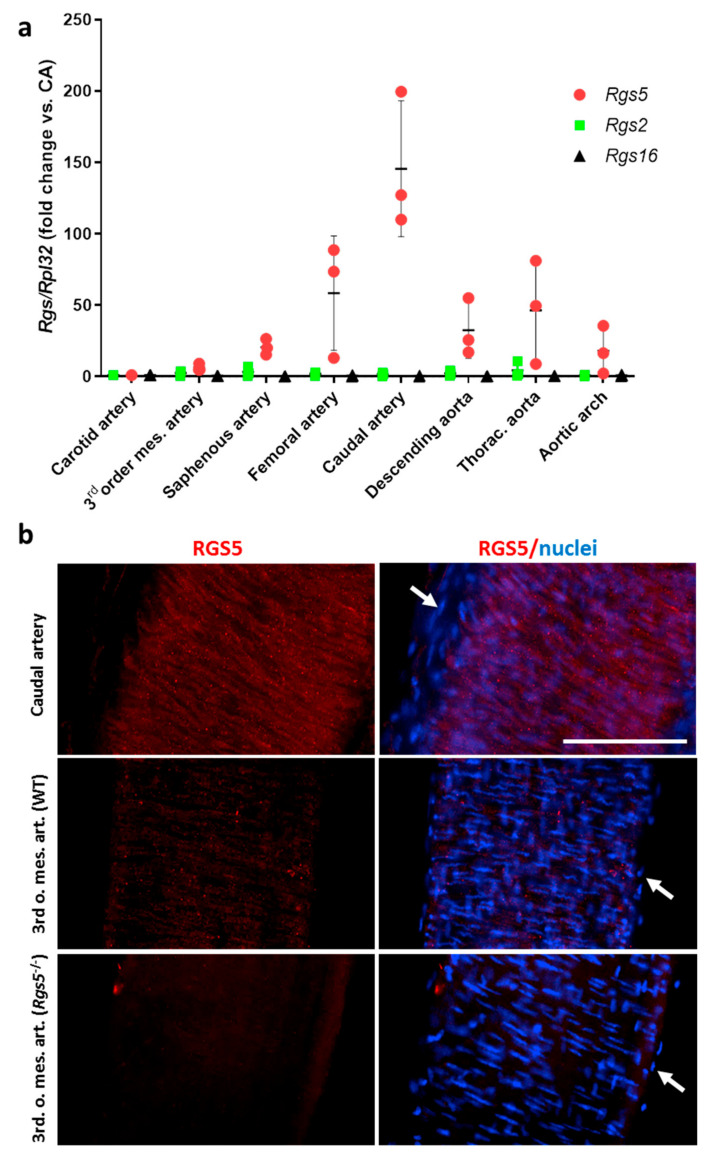
Analysis of the Rgs5 expression in different types of mouse arteries. (**a**) qPCR analyses were performed to determine the relative expression levels of *Rgs5, Rgs2,* and *Rgs16* in different arteries of adult C57 BL/6 mice as compared to the carotid artery as a reference (set to 1). Expression of the housekeeping gene *Rpl32* was used as internal reference (n = 3). Note that the overall expression levels of Rgs2 and Rgs16 are much lower as compared to *Rgs5*. (**b**) RGS5 was detected by whole mount immunofluorescence-based techniques (red fluorescence) in the caudal artery (upper panel) and 3rd order branches of the mesenteric artery of WT and *Rgs5^−/−^* mice. The images represent the maximum fluorescence intensity projection of a Z-stack generated by confocal microscopy including all layers of the corresponding arteries. VSMCs can by identified by their circular orientation (scale bar: 100 µm, arrows point at nuclei located in the adventitial layer).

**Figure 2 cells-10-01748-f002:**
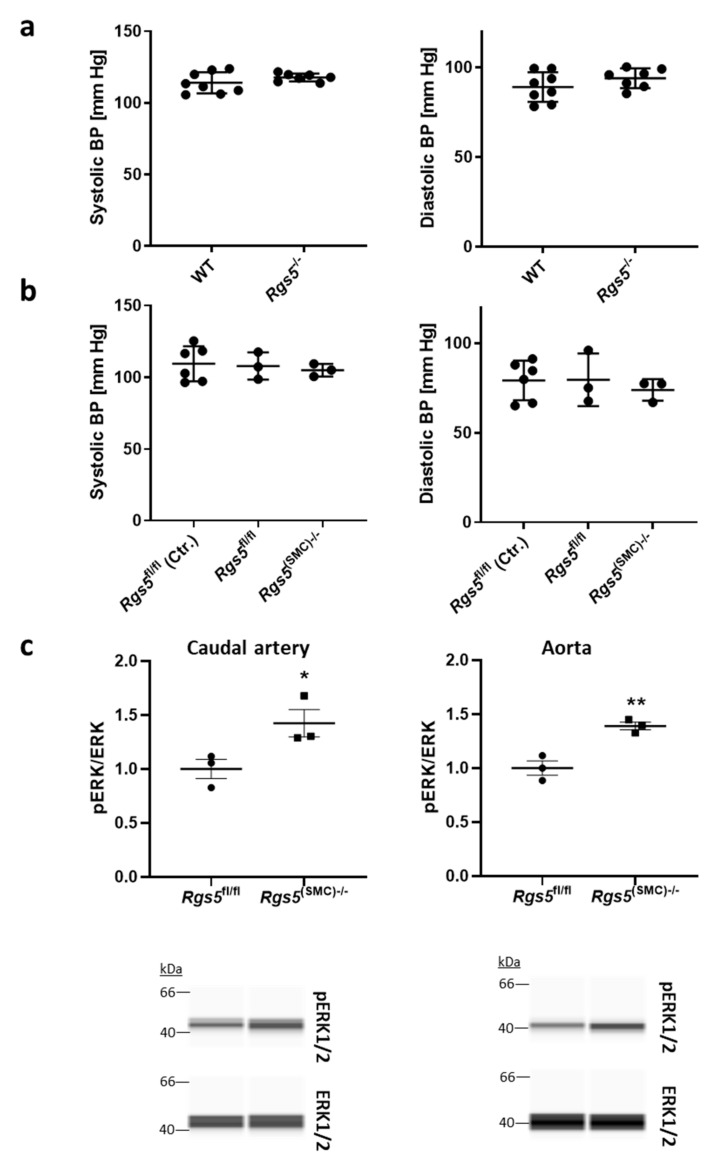
Telemetric assessment of the blood pressure and analysis of ERK1/2 phosphorylation in Rgs5 knockout mice. (**a**) The mean systolic and diastolic blood pressure were telemetrically recorded for 24 h in adult WT (n = 8) and *Rgs5*^−/−^ mice (n = 7). (**b**) Corresponding analyses were performed in SMMHC-CreER^T2^-*Rgs5*^fl/fl^ mice before (*Rgs5^fl/fl^* Ctr., n = 6) and 8 days after injection of Miglyol (*Rgs5*^fl/fl^, n = 3) or tamoxifen (*Rgs5*^(SMC)−/−^, n = 3) to induce the SMC-specific knockout of *Rgs5*. (**c**) Caudal arteries and aortas were collected from *Rgs5^fl/fl^* and *Rgs5^(SMC)−/−^* mice 3 weeks after induction of the knockout. pERK1/2 and total ERK1/2 abundance was quantified in tissue lysates by capillary electrophoresis/immunodetection-based techniques (* *p* < 0.05, ** *p* < 0.01, n = 3).

**Figure 3 cells-10-01748-f003:**
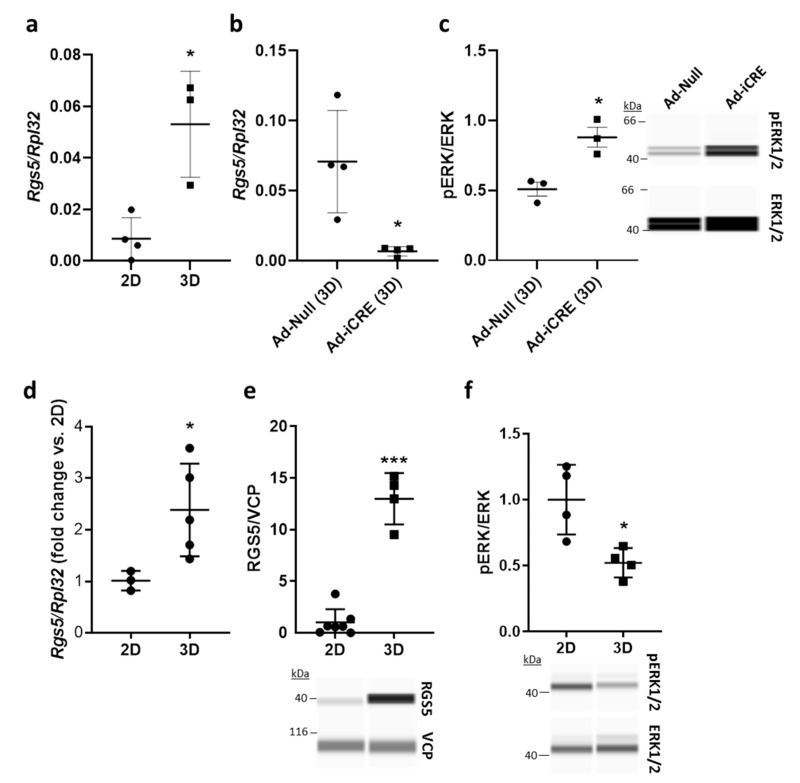
*VSMCs* cultured in *3D* spheroids display higher *RGS5* levels and low *ERK1/2* activity. (**a**–**c**) aoSMCs were isolated from the aortas of *Rgs5*^fl/fl^ mice and cultured as monolayer (2D) or spheroids (3D) for 48 h. (**a**) *Rgs5* expression was analyzed by qPCR with *Rpl32* as a reference (* *p* < 0.05, n = 3–4). (**b**) Knockout of *Rgs5* in aoSMCs was achieved by adenoviral transduction to overexpress Cre-recombinase (Ad-iCRE, control vector: Ad-Null) and verified by qPCR of RNA collected from 3D spheroids (* *p* < 0.05, n = 4). (**c**) Phosphorylation of ERK1/2 was significantly increased in 3D spheroids upon *Rgs5* knockout (* *p* < 0.05, n = 3). (**d**–**f**) HUASMCs were cultured as (2D) monolayer or 3D spheroids. (**d**) *RGS5* expression was determined by qPCR with *Rpl32* as internal reference (* *p* < 0.05, n = 3–5). (**e**) Capillary electrophoresis-based immunodetection of RGS5 and VCP as loading control (*** *p* < 0.001, n = 4–7) or (**f**) phospho-ERK1/2 and total ERK1/2 as reference (* *p* < 0.05, n = 4).

**Figure 4 cells-10-01748-f004:**
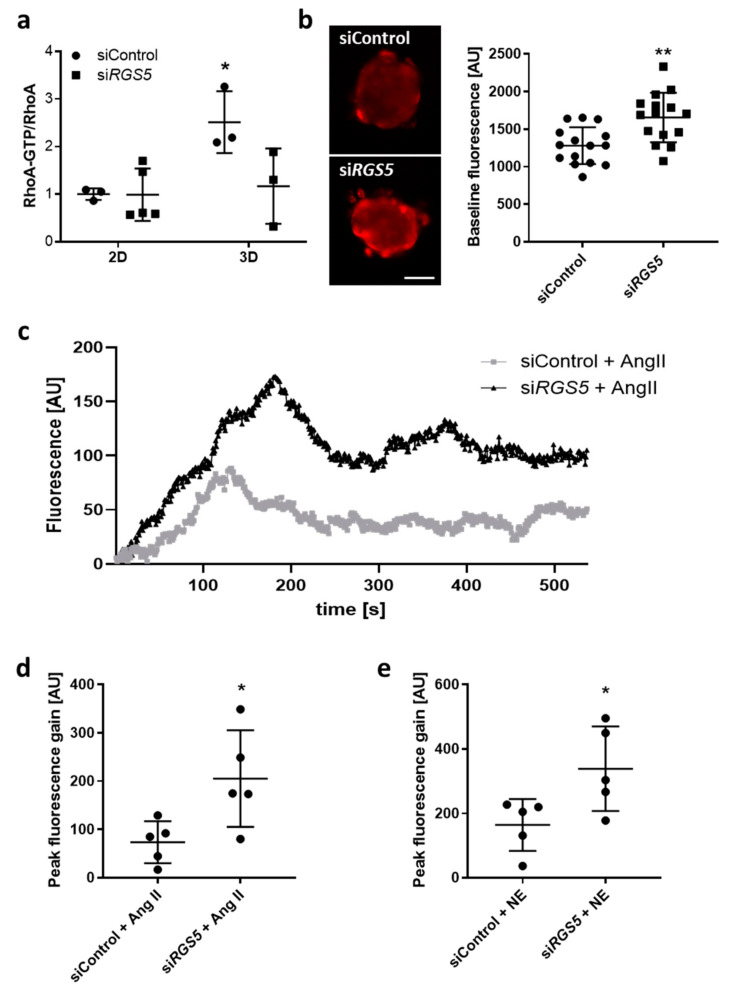
Loss *of RGS5* inhibits *RhoA*-activity and elevates calcium levels in *human VSMC* spheroids. RGS5 was depleted in HUASMC using siRNA-mediated transfection. Cells were further cultured as either monolayer (2D) or spheroids (3D). (**a**) Analysis of RhoA-GTP/RhoA applying a RhoA G-LISA. Total RhoA protein levels were determined using immunoblot-based techniques (* *p* < 0.05 vs. siControl, n = 3–5). (**b**–**e**) Three-dimensional (3D) spheroids were loaded with Rhodamin-4 to monitor the calcium-triggered change in fluorescence. (**b**) Representative image and corresponding quantification of the baseline calcium levels in HUASMC spheroids upon knockdown of *Rgs5* (** *p* < 0.01, results of 15 spheroids per group, scale bar: 50 µm). (**c**) Representative recordings of the angiotensin II (Ang II)-induced change of the fluorescence in spheroids upon knockdown of *Rgs5* vs. control conditions. (**d**,**e**) Corresponding quantification of the peak fluorescence gain (vs. baseline) in 3D spheroids (* *p* < 0.05, n = 5) upon treatment with Ang II or norepinephrine (NE).

**Figure 5 cells-10-01748-f005:**
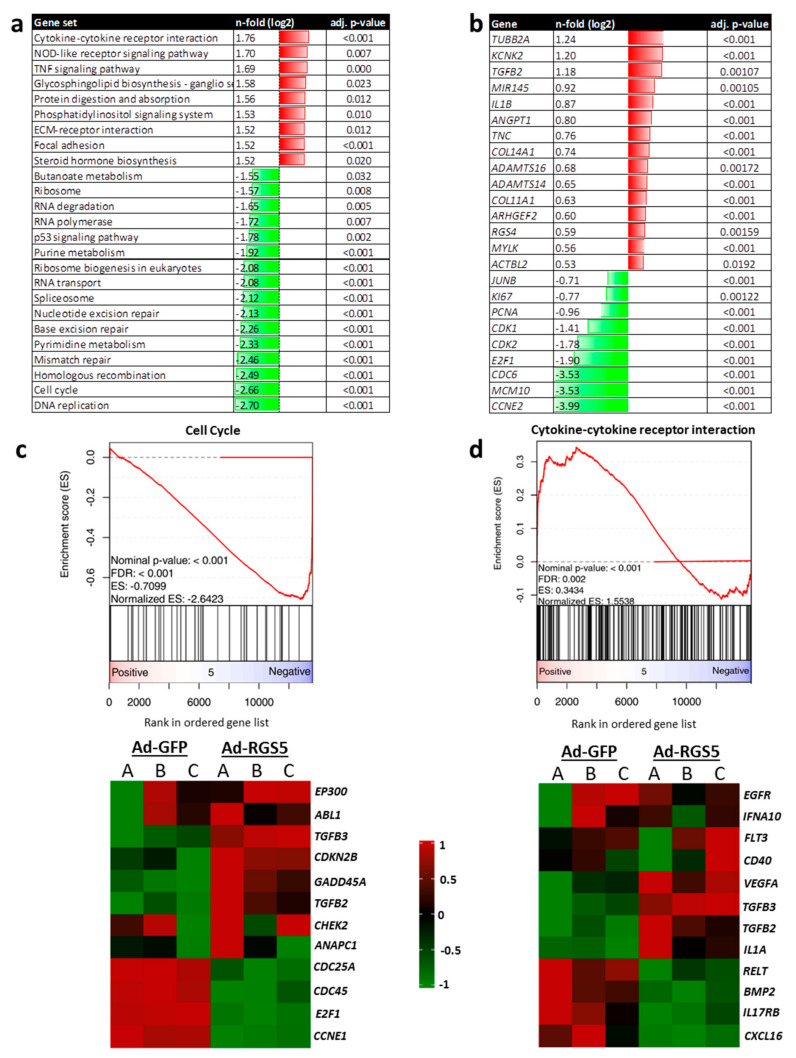
*RGS5* overexpression in human *VSMCs* induces cellular quiescence. RNA from HUASMCs overexpressing GFP (Ad-GFP) or RGS5 (Ad-RGS5) was subjected to transcriptome applying a whole genome microarray (n = 3). (**a**) GSE analyses were performed by clustering genes in sets from the KEGG database pathways. The table lists gene sets which were significantly up- (shown in red) and down-regulated (shown in green) upon *RGS5* overexpression with normalized enrichment score (NES) > 2.0/<−2.0 and adjusted *p*-value (adj. *p*) < 0.05. (**b**) List of individual top-regulated genes (red: upregulated, green: downregulated, RGS5 vs. GFP, selection criteria: log2-fold regulation >0.5/<−0.5; adjusted *p*-value (adj. *p*) < 0.05). (**c**,**d**) Representative enrichment score plots are shown for cell cycle- and cytokine-associated gene sets (NES: normalized enrichment score, adj. *p*: adjusted *p*-value). Heatmaps display the sample-specific regulation of exemplary genes attributed to these sets.

**Figure 6 cells-10-01748-f006:**
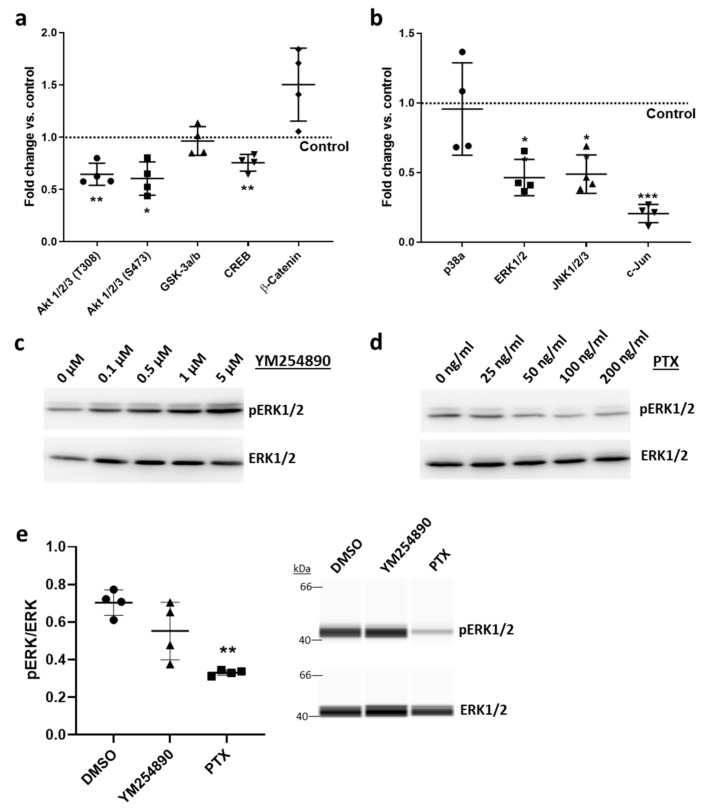
*RGS5* overexpression in human *VSMCs* inhibits the activity *of* several kinases. (**a**,**b**) The phosphorylation level of different protein kinases and some of their targets in HUASMCs overexpressing GFP (Control) or RGS5 was assessed by applying a protein kinase profiler array (* *p* < 0.05, ** *p* < 0.01, *** *p* < 0.001 vs. Control, n = 4). (**c**–**e**) Immunoblot-based analyses of ERK1/2 phosphorylation (pERK1/2) in HUASMCs treated with either Gα_q/11_ inhibitor (YM254890) or Gα_i/o_ inhibitor (Pertussis Toxin, PTX) for 24 h. (**c**,**d**) Concentration-dependent effect of YM254890 and PTX on ERK1/2 phosphorylation. (**e**) Capillary electrophoresis-based immunodetection of pERK1/2 and ERK1/2 in HUASMC after 24 h treatment with 2 µM of YM254890 and 200 ng/mL of PTX, respectively (** *p* < 0.01 vs. DMSO solvent control, n = 4).

**Figure 7 cells-10-01748-f007:**
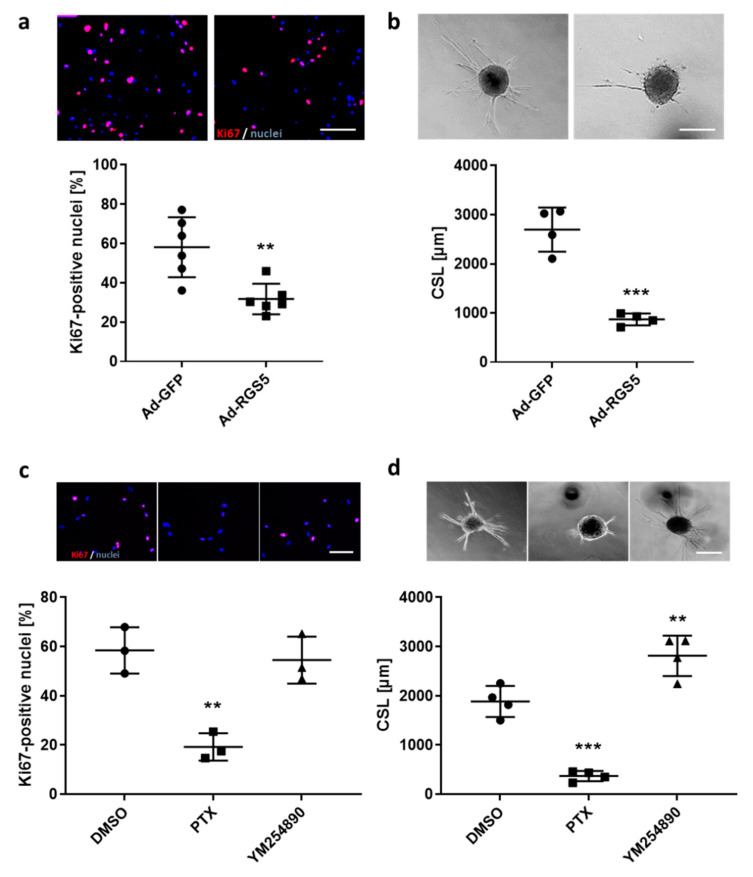
Overexpression of *RGS5* and inhibition *of* Gα_i/o_ limits proliferation and migration of human *VSMCs*. (**a**) Immunofluorescence-based detection of the proliferation marker Ki67 (Ki67: red, DAPI/nuclei: blue, scale bar: 200 μm) in HUASMC overexpressing GFP (Ad-GFP) or RGS5 (Ad-RGS5) after culture for 24 h (** *p* < 0.01, n = 6). (**b**) Analysis of HUASMC migration originating from 3D spheroids embedded in collagen gels for 24 h. The cumulative sprout length (CSL) was calculated by assessing the cumulative length of the sprouts from a single spheroid analyzing 10 spheroids per group and experiment (*** *p* < 0.001, n = 4, scale bar: 200 µm). (**c**,**d**) The aforementioned parameters were also quantified for HUASMCs treated with the Gα_i/o_ inhibitor (Pertussis Toxin, 200 ng/mL) or Gα_q/11_ inhibitor (YM254890, 5 µM) for 24 h (** *p* < 0.01, *** *p* < 0.001 vs. DMSO solvent control, n = 3/4).

**Table 1 cells-10-01748-t001:** PCR primer sequences.

Gene Name	Sequence	Annealing Temperature
***mRgs5***	5′-GCGGAGAAGGCAAAGCAA-3′ 5′-GTGGTCAATGTTCACCTCTTTAGG-3′	60 °C
***mRgs2***	5′-ATCAAGCCTTCTCCTGAGGAA-3′ 5′-GCCAGCAGTTCATCAAATGC-3′	
***mRgs16***	5′-CCTGGTACTTGCTACTCGCTTTT-3′ 5′-AGCACGTCGTGGAGAGGAT-3′	
***mRpl32***	5′-GGGAGCAACAAGAAAACCAA-3′ 5′-ATTGTGGACCAGGAACTTGC-3′	
***hRGS5***	5′-GGTGGAACCTTCCCTGAGCAGC-3′ 5′-AGAGCGCACAAAGCGAGGCA-3′	
***hRPL32***	5′-GTTCATCCGGCACCAGTCAG-3′ 5′-ACGTGCACATGAGCTGCCTAC-3′	

## Data Availability

The whole genome microarray data have been submitted to the Gene Expression Omnibus (GEO) repository. The records are available under GEO accession number GSE174130.

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
