# Peer review of "RGS5 Attenuates Baseline Activity of ERK1/2 and Promotes Growth Arrest of Vascular Smooth Muscle Cells"

_cells, 2021, doi:10.3390/cells10071748_

Round 1
Reviewer 1 Report
The work by Demirel et al. is an interesting study looking at the effect of RGS5 on signalling pathways in vascular smooth muscle cells. There is some interesting data here and lots of techniques but to me the rationale for some of the experiments isn't well explained and I'm not sure that the layout makes the most of the data. It would benefit from more coherent story telling as I feel just now it is presented more as "here is some data about RGS5 and here is some more data about RGS5" and I'm not sure of how you got from point a to b or why. There are so many different models and pathways to keep track of, it is quite complex.
Some concerns:
- I find the title confusing and think it should be simplified. Starting with "RGS5 channels..." makes me think that RGS5 are a type of ion channel that I haven't heard of before.
- The introduction has both too much and too little information in it. Too much background about GPCRs which is actually not that relevant to the data presented, but also too little information about the signalling pathways which are then investigated in the study. I found it confusing jumping about these different pathways without any grounding in how they get stimulated, and this is an area I am relatively comfortable with so I imagine very confusing for a more lay reader.
- Figure 1. Image quality is poor and could be improved, I'm not sure what I'm looking at. Whole mount staining should allow for staining in both EC and VSMC layers so it would be great to see both. I would also like to see it with relevant markers of each layer (e.g. alpha SM actin and PECAM). In the caudal artery the DAPI stain looks like it is not in the same cells are the RGS5 stain so I think this imaging needs to be more comprehensive either with better images or a different technique
- Figure 2. Can you show proof of global/SMC specific knockout? The rationale for looking at ERK1/2 phosphorylation is completely lost on me and needs to be explained more clearly especially as it forms the basis for the rest of the paper.
- Figure 4. You show that siRGS5 in your VSMC spheroids elevates calcium levels. So I'm wondering why you haven't looked at calcium levels in your RGS5 global/VSMC knockout mice? Is it altered here? There was no effect on basal BP but did you look response to AngII? Did you investigate ex vivo blood vessel contractility using force or pressure myography?
- Figure 5. Interesting data but I'm not sure how it links in with some of the other data. Did you do a microarray on your knockout mice? Does this correlate with what you see here?
Author Response
Please find our comments in the attached file.

Reviewer 2 Report
RGS5 Channels Intracellular Signaling Pathways to Foster Growth Arrest of Vascular Smooth Muscle Cells In the present study, the authors used loss- and gain-of-function studies to delineate the regulatory impact of RGS5 on mitogen-induced activation of the protein kinase ERK1/2 and associated functional responses of VSMCs. In my opinion, the study is interesting and innovative. However, I have some comments: Comment (1): Some parts of the text are confusing and needs a thorough revision of English. Comment (2): Abstract. The background topic is poor and there is not a clear statement about the aim of this study. I recommend to including a sentence about the main objectives of the study. Comment (3): Introduction. There is a brief review of existing knowledge and relevance of study. However, I would like to have some information about the protein kinase ERK1/2 as well in the introduction to understand the aim of the work. Comment (4): Materials and Methods. - Could you please write the city, state, country for each company from where you got your materials? - According to my experiences and doing the telemetric experiment several times, I am sure 2 months old mice are not in the good size to do the surgery to implant the devices. I would like to see a photo of the 2 months old mice. Comment (5): Results. A lot of information are included but there are some comments. - The first section “3.1.” without subtitle. - Again the first section “3.1.” should be moved to the introduction or even to the discussion. Comment (6): Discussion. Great information in this section but need a proofreading to improve it well. The authors concluded their works at the end systematically. Comment (7): Figures and legends. Legends are adequate and figures are necessary to understand the results obtained. - Authors need to provide a high-resolution version of Figure 5.Author Response
Please find our comments in the attachedf file.

Round 2
Reviewer 1 Report
I am happy with the paper in its revised form
Reviewer 2 Report
The Authors addressed all my previous comments in this improved version. I recommend to accept this MS in the present form.